# Do Mindful Eating and Intuitive Eating Affect Beverage Preferences? A Cross-Sectional Survey

**DOI:** 10.3390/foods13050646

**Published:** 2024-02-21

**Authors:** Gizem Kose, Murat Bas

**Affiliations:** Department of Nutrition and Dietetics, Faculty of Health Sciences, Acibadem Mehmet Ali Aydinlar University, 34752 Istanbul, Turkey; murat.bas@acibadem.edu.tr

**Keywords:** mindful eating, intuitive eating, beverages, sugar-sweetened beverages, alcoholic beverages

## Abstract

Intuitive eating involves following body signals to guide eating decisions and avoiding restrictive diets. Mindful eating is paying full attention to sensory experiences and fostering nonjudgmental awareness. We aimed to elucidate potential relationships between beverage intake and intuitive and mindful eating scores. This cross-sectional exploratory study (October 2021–December 2022) enrolled adult male and female participants who provided sociodemographic and health-related information and subsequently completed the Beverage Intake Questionnaire, the Mindful Eating Questionnaire (MEQ-30), and the Intuitive Eating Scale (IES-2). Bidirectional relationships were observed between beverage intake and intuitive and mindful eating scores. The total beverage intake of males was double compared with that of females, with the intake being mainly derived from sugar-sweetened beverages (*p* = 0.000). Increased total water intake was positively correlated with IES-2 and its subscale scores and was negatively correlated with MEQ-30 scores in both sexes (*p* < 0.05). Alcoholic beverage intake was associated with reductions in IES-2 and MEQ-30 scores, whereas unconditional permission to eat increased by 18.3% in males. In females, the consumption of milk-based beverages decreased the IES-2 scores. Understanding the complex relationships between beverage intake and intuitive and mindful eating may deepen our understanding of individual dietary behaviors and inform dietary interventions.

## 1. Introduction

Data from the World Health Organization (WHO) have shown that the prevalence of obesity continues to increase globally. Based on the WHO statistical data, the age-standardized prevalence of obesity among adults approximately tripled between 1975 and 2016, with 13.1% of adults worldwide being obese as of 2016. Furthermore, the highest age-standardized prevalence of obesity among adults has been observed in the American regions (28.6%), whereas the second-highest prevalence (23.3%) has been noted in the European region to which Turkiye belongs [1]. A previous examination of Turkiye-specific data revealed a 6.3% increase in obesity prevalence among females from 2008 to 2019. In 2019, the prevalence of obesity among females aged ≥15 years was 24.8%, whereas that among males was 17.3%. All above mentioned statistical data indicate that females have a higher obesity prevalence than males [2,3]. Increasing evidence has indicated the mechanisms underlying the associations between high sugar-sweetened beverage (SSB) intake and obesity. Although SSBs are energy-dense beverages with moderate-to-high glycemic index, their ability to increase satiety is limited; therefore, high SSB intakes may increase the risk of excessive weight gain, inflammation, insulin resistance, and atherogenic dyslipidemia [4]. Approximately 20% of total beverage intake is derived from liquids within foods; therefore, the adequate water intake should be at least 2000 mL/day [5]. The Institute of Medicine recommends identical amounts for euhydration maintenance [6]. Pehlivan et al. [7] conducted a study in Turkiye and reported that water and black tea were the most consumed beverages in this region. In their study, increased body mass index (BMI) was associated with a significantly decreased rate of insufficient fluid intake among the participants. The frequency of various chronic conditions, such as cardiovascular dysfunction, urinary tract infections, and circulation complications, was 4.2-fold higher in underhydrated adults than in euhydrated adults [8].

A fundamental indicator of our body’s need for hydration is the physical sensation of thirst. However, when selecting a beverage, external factors including the time of the day, recent food intake, availability, personal preferences, and financial considerations can influence the decision [9]. Public health initiatives focus on decreasing SSB intake while increasing water intake. Such strategies help limit sugar intake, improve weight management and hydration, and potentially mitigate diet-related health issues [10]. Sugar-added fruit juices and SSBs such as carbonated drinks, cola, and flavored sodas contribute to excessive energy intake [11]. Several pieces of evidence from experimental and observational studies have suggested that high SSB consumption contributes to excessive weight gain [12,13]. When eating occurs for reasons other than homeostatic regulation, the internal homeostatic signals regulating the initiation or termination of an eating event are either not released or are overridden [14]. Consuming food or beverages can be motivated externally. To reduce external motivations for eating, two types of interventions have been developed, namely, mindful eating (ME) and intuitive eating (IE). ME is defined as the nonjudgmental awareness of internal and external cues that influence the desire to eat, food choices, and consumption quality [15]. ME involves eating away from distractions, whereas IE emphasizes the significance of internal motivations related to the sensory properties of foods and internal cues of digestive behavior, such as physiological cues [16,17]. IE emphasizes trusting one’s body, fostering a positive relationship with food, and honoring the internal self rather than adhering to external recommendations [18]. IE is characterized by an association or alignment with internal cues of hunger and fullness.

Only two studies have delineated the relationships between IE subscales and general intake in a large adult population, and both studies have shown some differences based on sex; however, the relationships between IE and food items have shown mixed results. Camilleri et al. [19] reported that unconditional permission to eat was negatively associated with fruit, vegetable, and whole grain intakes, as well as positively correlated with sweet and fat intakes in both females and males. Results of published studies are contradictory as positive relationships between IE and diet quality [20,21,22,23] and no significant relationships [24,25]. As the relationships between IE and dietary intake may differ by sex and the fact that studies involving males are lacking, additional studies based on sex are warranted. On the other hand, individuals practicing ME exhibit several significant characteristics, including healthy dietary behaviors and a low prevalence of obesity, depression, and eating disorders [26,27]. ME involves making conscious food choices, developing awareness regarding food intake-related interoceptive cues, participating in cues of physical and psychological hunger, and appropriately responding to these cues [28]. ME may be particularly effective in altering the underlying regulatory processes of food intake associated with “liking” and “wanting” high-fat and sweet foods [29]. The Mindfulness-based Eating Awareness Training employed in the current intervention involves guiding individuals toward taste awareness, satiety, and sensation-specific satiety. This training aims to disrupt the tendency to overeat foods with high fat, sugar, and/or salt content while maintaining and even enhancing awareness regarding food preferences [28]. A relationship between unhealthy food choices and emotional and external nutrition has been previously shown in the research [30,31]. Another study reported that mindfulness was consistently associated with less impulsive eating, lower energy intake, and healthier snack choices. Therefore, eating mindfully may be a useful dietary strategy that positively influences food consumption for health promotion and disease prevention [32]. A previous systematic literature review reported improvements in targeted eating behaviors in 18 of 21 studies in which participants were subjected to mindfulness-based interventions [33].

There is ongoing interest in the potential health effects of beverage choices. Although the positive and negative effects of beverages such as orange juice and milk, which are classified as healthy drinks, have been reported, evidence has shown that tea and coffee positively affect cardiovascular health. Furthermore, the potential health risks associated with the consumption of beverages sweetened with various types of sugar, commonly referred to as SSBs, have attracted great interest [34]. To the best of our knowledge, original studies examining the impact of ME and IE behaviors on beverage preferences have not yet been reported in the literature. Therefore, this study aimed to assess the influence of ME and IE behaviors on beverage preferences.

## 2. Materials and Methods

### 2.1. Participants

Adults aged 22–65 years were enrolled in this study from October 2021 to December 2022. Pregnant or lactating females, participants with a history of psychiatric diseases, and those who self-reported regular administration of supplements or drugs were excluded. To achieve 90% power at a 95% confidence level, this study required a sample size of 385. Study power analysis was performed using the G* power 3.1.9.2 package.

### 2.2. Procedures

This cross-sectional exploratory study was conducted following the guidelines outlined in the Declaration of Helsinki. The Acibadem Healthcare Institutions Medical Research Ethics Committee approved the study protocol on 9 February 2021 (ref: 2021/16-30). Participants clicking on the “I give consent to take part in the research study” button on the information page were assumed to provide informed consent, which was approved by the committee and described as voluntary participation with anonymity. Once the participants consented, they could view and complete the online survey. We used snowball sampling for the present study. As all questions were mandatory, no data were lost. The first part of the questionnaire included 15 questions regarding sociodemographic factors, including age, sex, body weight, height, chronic disease history, and adherence to any special diets. After answering all demographic questions, the participants were asked to complete the following three questionnaires: Beverage Intake Questionnaire (BEVQ), Mindful Eating Questionnaire (MEQ-30), and Intuitive Eating Scale (IES-2).

### 2.3. Scales

#### 2.3.1. BEVQ

Hedrick et al. published the original BEVQ [35]. We added two culture-specific beverages (ayran and salep), and the final questionnaire assessed the consumption frequency of 21 different beverages on a Likert-type scale, with responses ranging from “never” to “more than three times a day.” Moreover, the participants were asked to estimate the amount of each beverage consumed per day in milliliters (mL). All non-water beverages were categorized as total, SSBs, sugar-free beverages, milk-based (containing milk protein), alcohol (e.g., beer and wine), and hard liquor. SSBs included carbonated, fruit, sports, energy, sweetened tea, and coffee drinks [36]. Sugar-free beverages included unsweetened tea and coffee, sparkling water (soda), and sugar-free light beverages. Any milk-containing beverage was classified as a milk-based beverage. Finally, alcoholic beverages were categorized as hard liquor, wine, and beer [6].

#### 2.3.2. MEQ-30

The MEQ is a 30-item scale developed by Framson et al. [37] and adapted by Gizem et al. [38] for use in Turkiye. The following seven subscales related to ME are rated on a Likert-type scale, with responses ranging from “never” to “always”: disinhibition (DH), emotional eating (EE), eating control (EC), eating discipline (ED), mindfulness (MF), conscious nutrition (CN), and interference (IF). Higher total and subscale scores indicate increased mindful eating.

#### 2.3.3. IES-2

The IES-2 is a 23-item 5-point Likert-type scale instrument that evaluates the four major components of IE: unconditional permission to eat (UPE) (6 items), eating for physical rather than emotional reasons (EPR) (8 items), reliance on hunger and satiety cues (RHSC) (6 items), and body–food choice congruence (B–FCC) (3 items) [39]. Bas et al. published that the Turkish form of IES-2 is reliable and valid for use in Turkiye [40]. Responses on this scale range from 1 (strongly disagree) to 5 (strongly agree). Higher scores correspond to increased intuitive eating levels.

### 2.4. Statistical Analysis

Categorical variables were presented as frequencies. The Shapiro–Wilk test was used to examine the conformity of numerical variables to a normal distribution. Normally distributed data were presented as means ± standard deviations ($\bar{X}$ ± SS), whereas non-normally distributed data were presented as medians (min–max). The difference between two independent groups that were not normally distributed was examined with the Mann–Whitney U Test. The relationship between two non-normally distributed quantitative variables was examined with Spearman’s rank difference correlation coefficient. To examine between-variable effects, logistic regression analysis was used. Statistical significance levels were reported at *p* < 0.05, *p* < 0.01, and *p* < 0.001; all interpretations were bidirectional. All study data were analyzed using the SPSS v26 (IBM Inc., Chicago, IL, USA) package program.

## 3. Results

A total of 1048 participants were enrolled in this study. Most males (*n* = 116, 46.8%) and females (*n* = 539, 67.4%) reported normal weight and had no chronic diseases (*n* = 892, 85.1%). Furthermore, 71.3% (*n* = 747) of the participants reported skipping main meals, and 48% (*n* = 503) of them engaged in regular physical activity (Table 1).

The intake of beverages, including water, vegetable juice, sugar-free coffee or tea, light milk, or buttermilk, showed no statistically significant differences between males and females (*p* > 0.05) (Table 2). The mean water (900 mL) intakes as well as ayran (buttermilk, 90 mL) intakes were similar between males and females. Overall, males consumed twice the amount of beverages as females (2161.84 ± 2312.03 vs. 994.12 ± 1314.16 mL; *p* = 0.000), most of which were SSBs, particularly sweetened tea and water. Females consumed lower amounts of all types of SSBs than males (471.29 ± 727.13 vs. 138.79 ± 288.92 mL; *p* = 0.000). Compared with females, males consumed significantly higher amounts of milk-based beverages (278.73 ± 496.78 vs. 186.00 ± 283.06 mL; *p* = 0.000) and sevenfold higher amounts of total alcoholic beverages (374.60 ± 833.62 vs. 53.74 ± 323.50 mL; *p* = 0.000), mainly wine and beer (Table 2).

Females had higher IES-2 and MEQ-30 scores than males (3.34 ± 0.71 and 2.90 ± 0.47 vs. 3.29 ± 0.77 and 2.74 ± 0.56, respectively). However, only MEQ-30 scores showed statistically significant differences (*p* = 0.000). Males had significantly higher subscale scores for UPE and EC than females (*p* = 0.000); however, females demonstrated significantly higher subscale scores for RHSC (*p* = 0.000), B–FCC (*p* = 0.000), EE (*p* = 0.000), EC (*p* = 0.000), MF (*p* = 0.003), and ED (*p* = 0.000) (Table 2).

As shown in Table 3, no correlations were noted between female (F) participants’ age, BMI, and IES-2 scores (*p* > 0.05). In contrast, males (M) showed a general decrease in IES-2 (r = −0.360) and its subscale scores with aging (all *p* = 0.000; UPE for males, *p* = 0.031). BMI was negatively correlated with IES-2 (F, r = −0.404; M, r = −0.328), UPE (F, r = −0.284; M, r = −0.209), EPR (F, r = −0.366; M, r = −0.330), RHSC (F, r = −0.315; M, r = −0.224), and B–FCC (F, r = −0.310; M, r = −0.320) scores (all *p* = 0.000; UPE for males, *p* = 0.001) (Table 3). Moreover, all types of beverages were correlated with IES-2 and its subscale scores to a certain extent. In both sexes, increases in total water intake were positively correlated with IES-2, UPE, EPR, RHSC, and B–FCC scores. Regarding total beverage intake, males showed a 15.5% increase in UPE scores, whereas a similar decrease in RHSC scores was observed in both sexes (*p* < 0.05); however, IES-2, EPR, and B–FCC scores showed no correlation. SSB intake was negatively correlated with decreases in IES-2, EPR, RHSC, and B–FCC scores by 18.8%, 14.1%, 26.9%, and 17.3% in females and 35.6%, 34.9%, 37.3%, and 30.5% in males, respectively (*p* = 0.000). Additionally, in females, UPE scores increased by 11.0% (*p* < 0.05). Regarding sugar-free beverages, IES-2, EPR, RHSC (*p* = 0.000), and B–FCC (*p* < 0.05) scores increased by approximately 20% in males, whereas females showed decreases in IES-2, UPE, and EPR subscale scores of 7.5%, 7.6%, and 8.4%, respectively (*p* < 0.05). The consumption of milk-based beverages, which are considered protein-based sources, resulted in 8.3%, 10.3%, and 8.7% decreases in IES-2, total, UPE, and RHSC scores of females, respectively; in males, only UPE scores increased by 13.8%.

In males, alcoholic beverage intake caused significant decreases in IES-2, EPR, RHSC (*p* = 0.000), and B–FCC (*p* < 0.05) scores by 24.1%, 24.6%, 30.2%, and 20%, respectively, whereas UPE scores increased by 18.3%. In females, alcoholic beverage intake was correlated with a 7.4% decrease in RHSC scores (*p* < 0.05). Similar results were observed for hard liquor, wine, and beer; however, wine intake was not correlated with UPE in males. In females, an increase in hard liquor intake was associated with an 8.3% decrease in B–FCC scores (*p* < 0.05) (Table 3).

Correlations between age, BMI, and various beverage types as well as ME total and subscale scores adjusted by sex are presented in Table 4. Age was negatively correlated with MEQ-30 scores (*p* < 0.05) in both sexes, with slight variations. Statistically significant negative correlations were observed between age and DH, EE, and IF in females (*p* < 0.01) but not in males. Age affects ED and CN, with ED and CN scores being decreased in males and increased in females with aging (*p* = 0.000; only ED in females, *p* < 0.05). As BMI increased, both males and females showed increases in EE and EC scores and decreases in ED scores (*p* < 0.05 in males, *p* = 0.000 in females). Notably, a minimum 25% increase in MEQ-30, DH, and CN scores and an 11.6% decrease in MF scores were noted in females (*p* = 0.000).

In males, total water intake caused increases in ED (*p* = 0.000), EC, and MF (*p* < 0.05) scores by 27.1%, 14.6%, and 13.2%, respectively. In females, MF (*p* = 0.000) and ED (*p* < 0.05) scores increased by 16.9% and 10.2%, respectively. Additionally, total water intake was negatively correlated with MEQ-30, IF (*p* < 0.05), DH, and EE (*p* = 0.000) scores at rates of 8.4%, 11.2%, 14.6%, and 20.9%, respectively. In both sexes, total beverage intake decreased ED scores; however, MF and CN scores decreased in males, whereas MEQ-30, DH, EE, and EC scores increased in females. SSB intake and an increase in IF scores were associated with similar results of MF, ED, and CN. Sugar-free beverage intake increased MEQ-30 and EC scores in both sexes and CN scores in males (*p* < 0.05). In females, milk-based beverage intake was positively correlated with most ME subscales (EE, EC, MF, and ED, *p* < 0.05) and total scores (*p* = 0.000); however, in males, only IF scores increased (*p* < 0.05).

With an increase in total alcoholic beverage intake, MEQ-30 scores, particularly DH, decreased in males but increased in females (*p* < 0.05). Moreover, this led to a decrease in ED, CN (*p* = 0.000), MF, and IF (*p* < 0.05) scores in males, whereas EC scores increased in females (*p* < 0.01). The examination of specific beverage types revealed that hard liquor and beer intakes were negatively correlated with MEQ-30, DH, MF, ED, CN, and IF scores in males and were positively correlated with DH and EC scores in females. In both sexes, hard liquor intake was associated with a decrease in MF scores (*p* = 0.000 in males, *p* < 0.05 in females). Increased wine intake was associated with an increase in EC scores in females (*p* < 0.05) (Table 4).

## 4. Discussion

To maintain water balance, the European Food Safety Authority recommends a daily water intake of 2500 and 2000 mL/day for adult males and females, respectively [5]. According to the 2017 Turkiye Nutrition and Health Survey, the mean daily water intake for individuals aged ≥15 years should be 1594.3 ± 968.99 (males, 1766.4 ± 1039.56; females, 1423.8 ± 860.38) mL [41]. Studies have reported that water and beverage intake can vary according to factors such as age, sex, and BMI [42,43]. Ferreira-Pêgo et al. investigated the total daily fluid intake of 16,276 adults from 13 different countries and revealed that approximately 60% of males and 50% of females did not consume the recommended amount of water [44]. In a study examining the total daily fluid intake habits of adults in Turkiye, the mean total daily fluid intake was 2270 mL/day (water, 1470 mL/day; other beverages, 800 mL/day). This study showed that older adult participants consumed healthy traditional beverages, including ayran, whereas younger participants preferred low-nutrient high-energy beverages [45]. In our study, no statistically significant difference was noted in beverage consumption between males and females for water, vegetable juice, unsweetened coffee or tea, light milk, and ayran. The mean water intakes were similar in males and females. Overall, males consumed twice the amount of beverages as females. Most of these drinks were SSBs, particularly sweetened tea and water. This study examined the associations between beverage intake and the ME and IE scores among the participants. Strong correlations and sex-adjusted differences were observed between beverage intake behaviors and total and subscale scores. Although healthy eating is imperative for well-being, beverages have special consideration. In particular, SSBs with a high glycemic load can disrupt hunger and satiety cues, thereby potentially leading to additional food intake [46]. In a study specifically examining sugar intake, participants exceeded the recommended daily limit of sugar intake with added sugars in food and beverages [47]. As a beverage, SSB can be considered a source of sugar but not water. Water intake enhances the body’s hunger and satiety cues [48]. Total water intake can facilitate regular eating and ME. In a cross-cultural study, Sims et al. reported that the mean daily water intake of Australians and Americans was 1796 and 1535 mL, respectively [49]. They noted that drinking 0.9 L of water daily affected all IE subscales, ED, and MF; however, no effects were observed on total ME. As this value is below the liquid requirements of Institute of Medicine [6], we believe that it negatively affected IF, DH, and EE in females in terms of appetite. Recently, adherence to the recommended liquid and water intakes has been low [8]. Although thirst is the primary indicator of the need to drink, external factors can influence beverage choice and intake [9]. In this study, the total beverage intake for males was approximately 2 L, which was twice as that for females. Most of this difference was attributed to SSB intake. The National Health and Nutrition Examination Survey (NHANES 2011–2016) showed similar results for water and total beverage intakes [36]. A population-based study showed similar results [50]; however, in our study, the water intake was approximately one-third lower in females. Regarding total beverage intake, ED, as well as hunger and satiety cues, were disrupted in both males and females. Additionally, in males, UPE scores increased and MF and CN scores decreased, whereas ME, DH, EE, and EC scores increased in females. Several sex-based differences were attributed to the beverage type.

Recent NHANES 2011–2016 data analyses have indicated an overall decline in SSB intake and an increase in plain drinking water intake [10]. Replacing caloric SSBs with water has become a priority for improving public health [36]. SSBs affect human metabolism in various ways. The primary biological mechanisms linking SSBs to weight gain include a reduced feeling of satiety following their intake compared with the intake of solid or protein-based foods and an insufficient compensatory decrease in overall energy intake [12]. The current study revealed that females consumed less SSBs, particularly sweetened tea and coffee, than males. Our results are consistent with those of previous population-based studies [49,50]. Another study reported that females consumed SSBs on most days or daily; however, additional details were not provided [51]. In our study, SSBs were negatively correlated with decreased frequency of IE and ME behaviors in both sexes, particularly in males, with only an increase in UPE scores in females. Camilleri et al. reported that the UPE scores of both males and females demonstrated a negative moderately strong correlation with SSB intake. ME, DH, and EE subscale scores were higher in males than in females [19]. ME was negatively correlated with daily energy, carbohydrate, and fat intakes [52]. ME was previously associated with healthy diet changes, such as consuming fewer sugary foods [53]. Mantzios et al. showed a decrease in sugar intake as the frequency of ME behaviors increased [54]. In a study assessing IE score quartiles, no significant between-quartile differences were observed for SSB intake; however, the top quartiles consumed more fruits and vegetables [55]. In a study examining sugar use in coffee, the ME intervention group was more successful than the control group in eliminating sugar and maintaining this change for at least 6 months [56]. Horwath et al. reported that high UPE subscale scores were positively correlated with higher SSB intake and lower vegetable and fruit intake [20]. In the NutriNet-Santé study, EPR showed inverse associations with SSBs [19].

A meta-analysis reported that high SSB and artificially sweetened beverage (ASB) intakes were significantly associated with obesity, diabetes, and hypertension. For each 250 mL daily increase in SSB or ASB intake, the risk of adverse health outcomes increased by at least 10% [57]. Another meta-analysis revealed that SSB intake was associated with a 9% increase in cardiovascular disease risk [58]. Females are more likely to consume at least one serving of low-calorie sweetened beverages daily than males. In contrast, both sexes are equally likely to consume one serving per day of water [59]. The current study showed that females consumed fewer sugar-free beverages than males. Moreover, sugar-free beverage intake was positively correlated with ME and EC in both sexes and with additional CN in males. IE scores of males increased by approximately 20%, whereas those of females decreased. In a nationwide study involving a random sample of middle-aged females, IE scores were unrelated to high-fat/high-sugar food or fruit intake [60].

Animal protein is crucial for health. Owing to the meal culture, milk-based beverages are essential protein sources in Turkiye [61]. The current study showed that males consumed more milk-based beverages than females. Ayran (buttermilk), which is prepared using yogurt, salt, and water, is a traditional beverage in Turkiye. We revealed that males and females consumed a mean amount of 90 mL of ayran daily. Milk-based beverage intake decreased IE scores in females; however, it increased UPE and most MEQ-30 subscale scores in males. A previous study on IE showed that B–FCC had small positive associations with dairy product intake in males [20]. Despite controversial results for both sexes, milk-based beverages are healthier than SSBs or sugar-free beverages. According to the American Heart Association, the influence of sugar substitutes on long-term health remains controversial [62]. Selecting water and animal-based beverages over SSBs can reduce sugar intake, increase protein intake, and help maintain adequate hydration [4].

Males consumed sevenfold higher amounts of alcoholic beverages than females (all types of alcoholic beverages, particularly wine and beer). These values remarkably exceeded those reported by Sims et al. [49], even considering the addition of alcoholic beverages. In the current study, alcoholic beverage intake in males led to undesirable decreases in IE and ME behaviors; however, UPE increased. Moreover, in females, alcoholic beverage intake appeared to disrupt RHSC, with an increase in ME and DH. When the intake of alcoholic beverages, including hard liquor, wine, and beer, was evaluated, similar results were obtained, with only wine intake showing no correlations with ME. These results suggest that alcoholic beverages are related to DH and lead to ignoring internal cues. Further correlations were noted in both sexes, particularly in males, possibly because of the high amount of alcohol intake. Alcohol intake is frequently reported as a coping mechanism for psychological factors, including alleviating stress [63]. However, during the coronavirus disease 2019 pandemic lockdowns, higher levels of psychological distress were associated with a higher intake of discretionary foods and SSBs but not alcohol [51]. Besides alcohol addiction, sugar intake has a more stringent relationship with stress. However, the factor that induced the alcohol intake levels in this study remains unclear.

Females had higher IE and ME scores than males, with ME scores showing a statistically significant difference. Regarding the subscale factors, males showed higher UPE and EC scores than females, whereas other subscale scores were generally higher in females. A positive correlation existed between ED and the healthy eating index and adherence to the Mediterranean diet mean scores. Males showed significantly higher mean scores for ME, DH, and EE subscales than females [52]. In another similar study, males scored higher on these subscales [20]. In both sexes, age was negatively correlated with ME, with slight differences. Statistically significant negative correlations were observed between age and DH, EE, and IF in females but not in males. As BMI values increased, EE and EC increased for both sexes, whereas IE (across all subscales) and ED decreased. EPR and relying on hunger/satiety cues showed moderate inverse correlations with external eating [64]. Conversely, in females, an increase in these factors caused at least a 20% increase in ME, DH, and CN scores. This finding may be linked to the lower prevalence of high BMI. External cues can differentiate BMI classifications, with normal-weight individuals showing an increased frequency of ME in food-related settings [65]. A strong correlation was observed between ME (but not MF) and BMI, indicating that individuals with healthier body weights were more likely to follow their internal eating experiences [66]. Several studies have reported relationships between BMI and ME-related factors, with lower BMI showing a significant association with ME-related scores [16,20,64,67].

In the current study, UPE, prompted by physical reasons and accompanied by RHSC, robustly affected daily nutrition, thereby leading to a 50% reduction in meal skipping. ED encompasses healthy eating, calorie intake awareness, and maintenance of regular mealtimes. In this study, individuals with greater ED were more likely to skip meals. This may be associated with hunger and satiety cues considering that a lack of these cues may lead to avoiding entire meals. Subtle body signals may guide healthier food choices and enhance well-being [54]. However, individuals with less bodily awareness may struggle to distinguish and interpret these signals, thereby possibly resulting in less healthy eating patterns. This highlights the need for improved awareness of ME and health-conscious food choices.

### Strengths, Limitations, and Future Research

The strengths of this study included the large random population sample, the inclusion of both sexes, and a thorough analysis of multiple dimensions of ME and IE; furthermore, we collected detailed information on beverage intake. To the best of our knowledge, no other studies have investigated the associations between beverage intake and ME. However, our results should be considered with caution owing to several study limitations, including the online survey design, which could have led to response errors as the participant pool was young, educated, and wealthy. Moreover, we only considered beverages in this study. Although the general public is becoming increasingly aware of the risks associated with SSB intake, there is little understanding of general beverage “healthfulness” [68]. By refocusing an individual’s attention on internal hunger cues and nutritional facts rather than external factors, ME may help reduce sugar-sweetened food consumption [16]. Additionally, future research should consider examining the effects of ME interventions for low-attention eaters.

## 5. Conclusions

This study enhances our understanding of how beverage consumption aligns with public health nutritional recommendations. Further, it supports that mindful eating and intuitive eating influence the beverage choices of several adults. These findings highlight that total water intake increases intuitive beverage choices and beverage types (particularly SSBs and alcoholic beverages) can disrupt internal cues for hunger cues. It is possible to make positive changes in the food and beverage preferences of individuals through training on mindful and intuitive eating. Thus, future studies evaluating the effectiveness of interventions using these behavioral models are crucial.

## Figures and Tables

**Table 1 foods-13-00646-t001:** Descriptive statistical data regarding body mass index, chronic diseases, smoking status, meal-skipping behavior, and physical activity according to sex (*n* = 1048).

	Sex					
	Males (*n* = 248)		Females (*n* = 800)		Total	
	*n*	%	*n*	%	*n*	%
Body mass index (BMI) classification						
Underweight (<18.5 kg/m^2^)	0	0.0	71	8.9	71	6.8
Normal weight (18.5–24.99 kg/m^2^)	116	46.8	539	67.4	655	62.5
Preobese (25–29.99 kg/m^2^)	105	42.3	143	17.9	248	23.7
Obese (≥30 kg/m^2^)	27	10.9	47	5.9	74	7.1
Chronic disease						
With concomitant disease(s)	33	13.3	123	15.4	156	14.9
No concomitant disease(s)	215	86.7	677	84.6	892	85.1
Disease-specific diet						
Follows a disease-specific diet	3	9.1	25	20.3	28	17.9
Does not follow a special diet	30	90.9	98	79.7	128	82.1
Smoking status						
Never smoker	71	28.6	476	59.5	547	52.2
Former smoker	36	14.5	140	17.5	176	16.8
Current smoker	141	56.9	184	23.0	325	31.0
Cigarettes (per day) ($\bar{X}\pm SS$)	20.39 ± 8.46		10.69 ± 7.42		14.90 ± 9.23	
Meal skipping behavior						
Skipping meals	166	66.9	581	72.6	747	71.3
Not skipping meals	82	33.1	219	27.4	301	28.7
Physical activity						
Regular physical activity	139	56.0	364	45.5	503	48.0
No regular physical activity	109	44.0	436	54.5	545	52.0

**Table 2 foods-13-00646-t002:** Age, body mass index, beverage consumption, IES-2, and MEQ-30 total and subscale scores according to sex.

	Males (*n* = 248)		Females (*n* = 800)			
	$\bar{X}\pm SS$	Median (Min–Max)	$\bar{X}\pm SS$	Median (Min–Max)	U	*p*
Age (years)	30.77 ± 11.32	25 (22–65)	32.27 ± 9.98	28 (18–64)	83,263	0.000 ***
BMI (kg/m^2^)	25.61 ± 3.52	25 (18.5–41)	23.08 ± 4.60	22.1 (15.4–54.2)	55,180	0.000 ***
Water (mL)	918.39 ± 395.02	720 (200–1500)	906.51 ± 406.39	720 (130–1500)	94,504	0.247
Total beverage intake (mL)	2161.84 ± 2312.03	1579 (0–21,499.2)	994.12 ± 1314.16	724.2 (0–28,500)	56,576	0.000 ***
Sugar-sweetened beverages (mL)	724.18 ± 980.95	288.3 (0–6240)	221.62 ± 475.41	33.6 (0–7500)	66,154	0.000 ***
Fruit juice (100%)	49.96 ± 154.67	0 (0–1500)	27.76 ± 105.05	0 (0–1500)	84,104.5	0.000 ***
Fruit juice	89.86 ± 297.03	0 (0–1500)	29.03 ± 103.03	0 (0–1500)	79,418	0.000 ***
Vegetable juice	27.85 ± 148.55	0 (0–1500)	16.61 ± 80.49	0 (0–1500)	98,587.5	0.803
Tea and coffee with sugar	471.29 ± 727.13	83.4 (0–3000)	138.79 ± 288.92	0 (0–3000)	79,958	0.000 ***
Energy drinks	23.68 ± 106.36	0 (0–1500)	4.54 ± 60.80	0 (0–1500)	82,674.5	0.000 ***
Carbonated drinks	179.25 ± 358.29	28 (0–1500)	50.53 ± 169.84	0 (0–1500)	70,353.5	0.000 ***
Sugar-free beverages (mL)	504.01 ± 518.56	390 (0–3000)	389.07 ± 405.96	332 (0–3000)	90,954.5	0.047 *
Coffee–tea (sugar-free)	432.25 ± 493.32	240 (0–1500)	351.85 ± 361.57	260 (0–1500)	97,348	0.653
Soda (sparkling water)	162.60 ± 323.99	71 (0–1500)	98.04 ± 170.64	33.6 (0–1500)	89,739	0.020 *
Light beverages	71.76 ± 207.60	0 (0–1500)	37.22 ± 156.87	0 (0–1500)	89,384	0.001 **
Milk-based beverages (mL)	278.73 ± 496.78	156.4 (0–3626.4)	186.00 ± 283.06	120 (0–6000)	89,463.5	0.019 *
Whole milk	85.45 ± 172.85	18.2 (0–1000)	37.09 ± 61.52	0 (0–600)	87,800	0.003 **
Semifat milk	40.55 ± 126.50	0 (0–1000)	39.03 ± 104.96	0 (0–1500)	91,828	0.042 *
Skim milk	34.02 ± 180.40	0 (0–1500)	12.28 ± 68.52	0 (0–1500)	98,823.5	0.873
Ayran (buttermilk)	98.63 ± 154.51	72 (0–1500)	92.79 ± 149.91	0 (0–1500)	96,540.5	0.517
Salep	20.08 ± 118.15	0 (0–1500)	4.82 ± 54.55	72 (0–1500)	91,044.5	0.000 ***
Alcoholic beverages (mL)	374.60 ± 833.62	67.2 (0–7500)	53.74 ± 323.50	0 (0–7500)	61,621	0.000 ***
Hard liquor	127.85 ± 341.59	0 (0–3000)	17.05 ± 146.86	0 (0–3000)	66,867	0.000 ***
Hard liquor (Whiskey–Raki)	65.96 ± 194.04	0 (0–1500)	9.06 ± 83.68	0 (0–1500)	71,442	0.000 ***
Liquor	61.88 ± 212.67	0 (0–1500)	8.00 ± 78.03	0 (0–1500)	78,144	0.000 ***
Wine	44.12 ± 181.76	0 (0–1500)	13.59 ± 95.19	0 (0–1500)	89,143	0.000 ***
Beer	202.64 ± 433.64	0 (0–3000)	23.10 ± 129.57	0 (0–3000)	62,617	0.000 ***
Light beer	64.23 ± 189.32	0 (0–1500)	9.22 ± 65.80	0 (0–1500)	79,929	0.000 ***
Beer	138.41 ± 328.02	0 (0–1500)	13.88 ± 72.66	0 (0–1500)	62,328.5	0.000 ***
Total IES-2 scores	3.29 ± 0.77	3.4 (1.7–4.7)	3.34 ± 0.71	3.3 (1.4–4.9)	95,353.5	0.356
UPE	3.57 ± 0.60	3.6 (2–5)	3.23 ± 0.70	3.2 (1.4–5)	70,808	0.000 ***
EPR	3.34 ± 0.83	3.3 (1–4.9)	3.26 ± 0.90	3.3 (1–5)	94,400	0.249
RHSC	3.11 ± 1.17	3.3 (1–5)	3.52 ± 0.88	3.7 (1–5)	82,070.5	0.000 ***
B–FCC	2.92 ± 1.44	3 (1–5)	3.38 ± 1.22	3.5 (1–5)	80,821.5	0.000 ***
Total MEQ-30 scores	2.74 ± 0.56	2.8 (1.6–4.5)	2.90 ± 0.47	2.9 (1.6–5)	85,266	0.001 **
Disinhibition	2.63 ± 0.92	2.4 (1–5)	2.64 ± 0.90	2.6 (1–5)	98,300.5	0.829
Emotional eating	2.24 ± 0.98	2 (1–5)	2.64 ± 1.10	2.6 (1–5)	78,204	0.000 ***
Eating control	2.64 ± 0.710	2.5 (1.3–4.3)	2.48 ± 0.66	2.3 (1–5)	84,618	0.000 ***
Mindfulness	3.64 ± 0.90	3.8 (1.4–5)	3.88 ± 0.69	4 (1.8–5)	86,859	0.003 **
Eating discipline	2.95 ± 0.85	3 (1–5)	3.39 ± 0.75	3.5 (1.5–5)	66,783	0.000 ***
Conscious nutrition	2.50 ± 0.59	2.4 (1.2–4)	2.57 ± 0.60	2.6 (1–5)	91,955.5	0.080
Interference	2.48 ± 0.82	2.5 (1–5)	2.49 ± 0.92	2.5 (1–5)	98,914.5	0.945

Abbreviations: * indicates a significant difference; * *p* < 0.05; ** *p* < 0.01; *** *p* < 0.001. BMI, body mass index; IES-2, Intuitive Eating Scale; UPE, unconditional permission to eat; EPR, eating for physical rather than emotional reasons; RHSC, reliance on hunger and satiety cues; B–FCC, body–food choice congruence; MEQ-30, Mindful Eating Questionnaire; U, Mann–Whitney U test.

**Table 3 foods-13-00646-t003:** Correlations between age, body mass index, total water intake, classified beverage intake, and IES-2 subscale scores with total scores according to sex.

		Males					Females				
		IES-2	UPE	EPR	RHSC	B–FCC	IES-2	UPE	EPR	RHSC	B–FCC
Age (years)	s	−0.360	−0.137	−0.325	−0.265	−0.414	−0.008	−0.037	0.047	−0.011	−0.052
	*p*	0.000 ***	0.031 *	0.000 ***	0.000 ***	0.000 ***	0.831	0.291	0.186	0.758	0.144
BMI (kg/m^2^)	s	−0.328	−0.209	−0.330	−0.224	−0.320	−0.404	−0.284	−0.366	−0.315	−0.310
	*p*	0.000 ***	0.001 **	0.000 ***	0.000 ***	0.000 ***	0.000 ***	0.000 ***	0.000 ***	0.000 ***	0.000 ***
Total water (mL)	s	0.390	0.206	0.334	0.380	0.338	0.220	0.171	0.196	0.207	0.164
	*p*	0.000 ***	0.001 **	0.000 ***	0.000 ***	0.000 ***	0.000 ***	0.000 ***	0.000 ***	0.000 ***	0.000 ***
Total beverage intake (mL)	s	−0.119	0.155	−0.116	−0.166	−0.080	−0.163	−0.026	−0.156	−0.153	−0.101
	*p*	0.062	0.014 *	0.068	0.009 **	0.211	0.000 ***	0.467	0.000 ***	0.000 ***	0.004 **
SSBs	s	−0.356	0.072	−0.349	−0.373	−0.305	−0.188	0.110	−0.141	−0.269	−0.173
	*p*	0.000 ***	0.256	0.000 ***	0.000 ***	0.000 ***	0.000 ***	0.002 **	0.000 ***	0.000 ***	0.000 ***
Sugar-free beverages (mL)	s	0.220	0.045	0.226	0.217	0.197	−0.075	−0.076	−0.084	−0.034	−0.045
	*p*	0.000 ***	0.482	0.000 ***	0.001 **	0.002 **	0.034 *	0.032 *	0.017 *	0.332	0.207
Milk-based beverages (mL)	s	0.019	0.138	−0.009	−0.013	0.085	−0.083	−0.103	−0.054	−0.087	0.030
	*p*	0.765	0.030 *	0.889	0.837	0.181	0.019 *	0.003 **	0.128	0.014 *	0.394
Alcoholic beverages (mL)	s	−0.241	0.183	−0.246	−0.302	−0.200	−0.027	−0.025	−0.019	−0.074	−0.011
	*p*	0.000 ***	0.004 **	0.000 ***	0.000 ***	0.002 **	0.452	0.487	0.600	0.037 *	0.762
Hard liquor (mL)	s	−0.231	0.165	−0.202	−0.310	−0.215	−0.045	−0.034	−0.039	−0.057	−0.083
	*p*	0.000 ***	0.009 **	0.001 **	0.000 ***	0.001 **	0.206	0.331	0.269	0.108	0.019 *
Wine (mL)	s	−0.195	0.083	−0.192	−0.225	−0.152	0.015	−0.005	0.035	−0.042	0.019
	*p*	0.002 **	0.192	0.002 **	0.000 ***	0.016 *	0.679	0.894	0.326	0.234	0.583
Beer (mL)	s	−0.176	0.205	−0.184	−0.245	−0.137	−0.018	−0.015	−0.020	−0.040	−0.014
	*p*	0.005 **	0.001 **	0.004 **	0.000 ***	0.031 *	0.608	0.662	0.579	0.256	0.684

Abbreviations: * indicates a significant difference; * *p* < 0.05; ** *p* < 0.01; *** *p* < 0.001. BMI, body mass index; s, Spearman’s rank correlation coefficient; IES-2, Intuitive Eating Scale; UPE, unconditional permission to eat; EPR, eating for physical rather than emotional reasons; RHSC, reliance on hunger and satiety cues; B–FCC, body–food choice congruence.

**Table 4 foods-13-00646-t004:** Correlations between age, body mass index, total water, classified beverage consumptions, and MEQ-30 subscale scores with total scores according to sex.

		Males								Females							
		MEQ	DH	EE	EC	MF	ED	CN	IF	MEQ	DH	EE	EC	MF	ED	CN	IF
Age (years)	s	−0.133	−0.109	−0.178	0.028	0.019	0.070	0.133	−0.280	−0.082	−0.025	−0.017	−0.063	−0.275	−0.222	−0.231	−0.052
	*p*	0.037 *	0.002 **	0.000 ***	0.432	0.583	0.046 *	0.000 ***	0.000 ***	0.020 *	0.691	0.795	0.327	0.000 ***	0.000 ***	0.000 ***	0.413
BMI (kg/m^2^)	s	0.095	0.270	0.208	0.271	−0.116	−0.131	0.251	0.040	0.237	0.049	0.157	0.187	−0.081	−0.153	0.048	0.037
	*p*	0.136	0.000 ***	0.000 ***	0.000 ***	0.001 **	0.000 ***	0.000 ***	0.260	0.000 ***	0.441	0.013 *	0.003 **	0.204	0.016 *	0.449	0.566
Total water (mL)	s	0.069	−0.146	−0.209	−0.064	0.169	0.102	0.061	−0.112	−0.084	−0.014	−0.110	0.146	0.132	0.271	0.090	0.021
	*p*	0.280	0.000 ***	0.000 ***	0.070	0.000 ***	0.004 **	0.087	0.002 **	0.017 *	0.823	0.085	0.022 *	0.038 *	0.000 ***	0.158	0.741
Total beverage intake (mL)	s	−0.039	0.128	0.117	0.150	−0.011	−0.087	0.023	0.039	0.132	0.042	0.012	0.068	−0.151	−0.248	−0.132	0.000
	*p*	0.537	0.000 ***	0.001 **	0.000 ***	0.762	0.014 *	0.511	0.277	0.000 ***	0.507	0.850	0.284	0.018 *	0.000 ***	0.038 *	0.998
SSBs	s	−0.084	0.115	0.114	0.001	−0.143	−0.304	−0.104	0.208	−0.006	0.059	0.098	−0.014	−0.183	−0.407	−0.159	0.154
	*p*	0.188	0.001 **	0.001 **	0.974	0.000 ***	0.000 ***	0.003 **	0.000 ***	0.875	0.355	0.123	0.821	0.004 **	0.000 ***	0.012 *	0.015 *
Sugar-free beverages (mL)	s	0.160	0.047	0.067	0.101	0.019	0.047	0.047	−0.064	0.095	0.094	0.024	0.148	0.096	0.060	0.156	0.048
	*p*	0.012 *	0.181	0.058	0.004 **	0.591	0.182	0.188	0.071	0.007 **	0.141	0.712	0.020 *	0.132	0.349	0.014 *	0.451
Milk-based beverages (mL)	s	0.010	0.065	0.084	0.080	0.095	0.095	0.042	0.041	0.138	0.095	0.041	0.026	−0.056	−0.088	−0.054	−0.137
	*p*	0.881	0.066	0.017 *	0.023 *	0.007 **	0.007 **	0.233	0.251	0.000 ***	0.136	0.518	0.681	0.379	0.167	0.397	0.031 *
Alcoholic beverages (mL)	s	−0.194	0.089	0.059	0.102	0.060	0.039	0.021	0.046	0.120	−0.166	−0.045	−0.026	−0.185	−0.316	−0.227	−0.153
	*p*	0.002 **	0.012 *	0.093	0.004 **	0.090	0.275	0.548	0.190	0.001 **	0.009 **	0.480	0.684	0.003 **	0.000 ***	0.000 ***	0.016 *
Hard liquor (mL)	s	−0.178	0.082	0.034	0.077	−0.070	−0.049	0.033	0.059	0.056	−0.186	−0.014	−0.069	−0.152	−0.271	−0.180	−0.165
	*p*	0.005 **	0.021 *	0.332	0.029 *	0.047 *	0.165	0.347	0.095	0.114	0.003 **	0.831	0.277	0.017 *	0.000 ***	0.004 **	0.009 **
Wine (mL)	s	−0.058	0.040	−0.005	0.086	0.033	0.056	0.004	0.027	0.054	−0.055	0.078	−0.015	−0.160	−0.106	−0.107	0.000
	*p*	0.364	0.256	0.893	0.015 *	0.353	0.112	0.911	0.443	0.124	0.390	0.219	0.809	0.012 *	0.096	0.093	1.000
Beer (mL)	s	−0.197	0.087	0.066	0.089	0.015	−0.002	−0.020	0.041	0.086	−0.176	−0.067	−0.026	−0.185	−0.276	−0.194	−0.165
	*p*	0.002 **	0.014 *	0.061	0.012 *	0.669	0.953	0.570	0.252	0.015 *	0.005 **	0.290	0.687	0.004 **	0.000 ***	0.002 **	0.009 **

Abbreviations: * indicates a significant difference; * *p* < 0.05; ** *p* < 0.01; *** *p* < 0.001. BMI, body mass index; s, Spearman’s rank correlation coefficient; MEQ-30, Mindful Eating Questionnaire-30; DH, disinhibition; EE, emotional eating; EC, eating control; ED, eating discipline; MF, mindfulness; CN, conscious nutrition; IF, interference.

## Data Availability

The original contributions presented in the study are included in the Appendix A, further inquiries can be directed to the corresponding author.

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
