# Peer review of "Do Mindful Eating and Intuitive Eating Affect Beverage Preferences? A Cross-Sectional Survey"

_foods, 2024, doi:10.3390/foods13050646_

Round 1
Reviewer 1 Report
Comments and Suggestions for Authors
I am pleased to review this article, which presents a thorough examination of mindful eating, intuitive eating, beverage choices, and their impacts on health outcomes. The authors have meticulously analyzed various relationships among these variables and discussed their implications in the context of existing literature. This paper has the potential to make a significant contribution to the field, especially if the authors address certain areas for improvement.
- The introduction is overly long and could be broken into two sections - a condensed introduction followed by a separate theoretical framework section going into more detail. This would improve readability.
- The results section is complex and difficult to digest. Consider simplifying the presentation of results, perhaps moving some detailed analyses to an appendix or supplementary materials section. Using more tables and figures to visualize key findings could also increase clarity.
- A conceptual model illustrating the hypothesized relationships between the key variables of mindful eating, intuitive eating, beverage choices, and health outcomes is lacking. Including such a model early on would orient the reader.
- The sample lacks diversity on key demographic factors like age, education, and income. Comment on how this may limit generalizability of findings. Consider undersampling to improve representation.
- Further justify the choice of statistical tests used to examine relationships between non-normal variables. Spearman correlations may not be optimal in all cases.
- The discussion section focuses heavily on comparing results to past studies. More interpretation of the meaning and implications of the novel findings on beverage choices is needed.
- Proofread the manuscript carefully - there are minor typos, formatting inconsistencies, and stylistic issues throughout that need addressing. Careful editing would polish the work.
- Supplement analyses by sex with tests of statistical interaction effects between key variables and sex. This would strengthen conclusions about differences in relationships by sex. Consider structural equation modeling to test multivariate relationships.
Author Response
Reviewer 1
I am pleased to review this article, which presents a thorough examination of mindful eating, intuitive eating, beverage choices, and their impacts on health outcomes. The authors have meticulously analyzed various relationships among these variables and discussed their implications in the context of existing literature. This paper has the potential to make a significant contribution to the field, especially if the authors address certain areas for improvement.
Thank you for your valuable time and important comments.
- The introduction is overly long and could be broken into two sections - a condensed introduction followed by a separate theoretical framework section going into more detail. This would improve readability.
Response and Revisions 1: We revised the introduction section and edited our manuscript with a professional editing service.
- The results section is complex and difficult to digest. Consider simplifying the presentation of results, perhaps moving some detailed analyses to an appendix or supplementary materials section. Using more tables and figures to visualize key findings could also increase clarity.
Response and Revisions 2: We revised the results section.
- A conceptual model illustrating the hypothesized relationships between the key variables of mindful eating, intuitive eating, beverage choices, and health outcomes is lacking. Including such a model early on would orient the reader.
Response and Revisions 3: We made a conceptual model in line with your suggestion. You can find them at the end of the manuscript. If you recommend, we can add them into our manuscript.
- The sample lacks diversity on key demographic factors like age, education, and income. Comment on how this may limit generalizability of findings. Consider undersampling to improve representation.
Response and Revisions 4: Thank you for your suggestions, we added this information as a limitation.
- Further justify the choice of statistical tests used to examine relationships between non-normal variables. Spearman correlations may not be optimal in all cases.
Response and Revisions 5: We used Spearman correlations due to our non-normal variables. We added this information as ‘The difference between two independent groups that were not normally distributed was examined with the Mann-Whitney U Test. The relationship between two non-normally distributed quantitative variables was examined with Spearman's rank difference correlation coefficient.’
- The discussion section focuses heavily on comparing results to past studies. More interpretation of the meaning and implications of the novel findings on beverage choices is needed.
Response and Revisions 6: We could not understand your suggestion. But if you can give an example, we are ready to implement your suggestions.
- Proofread the manuscript carefully - there are minor typos, formatting inconsistencies, and stylistic issues throughout that need addressing. Careful editing would polish the work.
Response and Revisions 7: Thank you for your suggestion. We edited our manuscript with a professional editing service.
- Supplement analyses by sex with tests of statistical interaction effects between key variables and sex. This would strengthen conclusions about differences in relationships by sex. Consider structural equation modeling to test multivariate relationships.
Response and Revisions 8: Thank you for your suggestion. Our aim was reveal beverage intake relations with intuitive and mindful eating but of course we would like to consider your suggestion for our future research.
Reviewer 2 Report
Comments and Suggestions for Authors
There are a lot of typesetting and formatting problems as well as language problems. The results and discussion are more like a report than a scientific paper. Therefore, I think this article is not suitable in it's current state.
Comments on the Quality of English Language
The language needs to be proofread by an English professional.
Author Response
Reviewer 2
Comments and Suggestions for Authors
There are a lot of typesetting and formatting problems as well as language problems.
Thank you for your valuable time. We edited our manuscript with a professional editing service.
- The results and discussion are more like a report than a scientific paper. Therefore, I think this article is not suitable in it's current state.
Response and Revisions 1: We revised the manuscript totally.
Comments on the Quality of English Language
- The language needs to be proofread by an English professional.
Response and Revisions 2: We edited our manuscript with a professional editing service.
Reviewer 3 Report
Comments and Suggestions for Authors
The study focuses on the role of food and the choices involved in the process, pointing to the importance of conscious eating. It also relates to possible relationships in beverage selection. Bidirectional relationships between beverage intake and intuitive and conscious eating were observed. The importance of the understanding of individual dietary behaviors under a complex relationship between beverage intake and intuitive and conscious eating is highlighted. The introduction gives a comprehensive and detailed overview of the evolution of food and beverage intake and its associated regulations and their scope. The materials and methods provide a comprehensive overview in detail and their statistical analysis. The results are well-defined in tables that clearly show the relevant findings, followed by a complete and precise discussion. In the conclusions, the significant findings of this research are presented, leading to recommendations for better public health. It underlines the fundamental role of food awareness and its influence on beverage choices. At the same time, it also shows the possibility of positive changes in food and beverage preferences. It is an important, relevant study that provides information for designing strategies to improve the public health sector.
Author Response
Response and Revisions 1: Thank you for your comments.
Also, we edited our manuscript with a professional editing service.
Reviewer 4 Report
Comments and Suggestions for Authors
This paper is well written and organized. However, there are several limitations before further consideration.
1. Introduction is too long to understand the key contributions. I recommend that the authors should reduce it and strengthen its selling points. Especially, I not sure that the key theoretical contribution are.
2. To this end, there no no theoretical framework. I recommend that the authors add theoretical background after the introduction.
3. The description of data collection is unclear. What are the samle selection criteria?
4. How did you solve the response and common method bias?
Author Response
Reviewer 4
This paper is well written and organized. However, there are several limitations before further consideration.
Thank you for your valuable time and important comments.
- Introduction is too long to understand the key contributions. I recommend that the authors should reduce it and strengthen its selling points. Especially, I not sure that the key theoretical contribution are.
Response and Revisions 1: We revised the introduction section and edited our manuscript with a professional editing service.
- To this end, there no no theoretical framework. I recommend that the authors add theoretical background after the introduction.
Response and Revisions 2: We made 2 conceptual model in line with your suggestion. You can find them at the end of the document. If you recommend, we can add them into our manuscript.
- The description of data collection is unclear. What are the samle selection criteria?
Response and Revisions 3: We used snowball sampling and added to the method section.
- How did you solve the response and common method bias?
Response and Revisions 4: We focused on behaviors as a partial solution. We provided information about confidentiality to have reliable answers. And also, we used validated scales as two attuned eating related scales with reverse coded.
Round 2
Reviewer 1 Report
Comments and Suggestions for Authors
I have no further comments.
Author Response
Dear Reviewer,
Thank you very much for your valuable time.
Kind Regards,
Reviewer 2 Report
Comments and Suggestions for Authors
Section 5 conclusion. The conclusion lacks data support, and important data needs to be mentioned in the conclusion.
Author Response
Dear Reviewer,
We revised conclusion in line with your suggestion.
Reviewer 4 Report
Comments and Suggestions for Authors
The revision is acceptable.
Author Response

(The authors gave the same response as above.)
